# Antiplasticization of Polymer Materials: Structural Aspects and Effects on Mechanical and Diffusion-Controlled Properties

**DOI:** 10.3390/polym12040769

**Published:** 2020-04-01

**Authors:** Leno Mascia, Yannis Kouparitsas, Davide Nocita, Xujin Bao

**Affiliations:** 1Department of Materials, Loughborough University, Loughborough LE11 3TU, UK; i.kouparitsas@gmail.com (Y.K.); x.bao@lboro.ac.uk (X.B.); 2Polymer IRC, Faculty of Engineering and Informatics, University of Bradford, Bradford BD7 1DP, UK; d.nocita@bradford.ac.uk

**Keywords:** antiplasticization, carbohydrate, glassy polymers, membrane, pharmaceutical, physical ageing, plasticizer

## Abstract

Antiplasticization of glassy polymers, arising from the addition of small amounts of plasticizer, was examined to highlight the developments that have taken place over the last few decades, aiming to fill gaps of knowledge in the large number of disjointed publications. The analysis includes the role of polymer/plasticizer molecular interactions and the conditions leading to the cross-over from antiplasticization to plasticization. This was based on molecular dynamics considerations of thermal transitions and related relaxation spectra, alongside the deviation of free volumes from the additivity rule. Useful insights were gained from an analysis of data on molecular glasses, including the implications of the glass fragility concept. The effects of molecular packing resulting from antiplasticization are also discussed in the context of physical ageing. These include considerations on the effects on mechanical properties and diffusion-controlled behaviour. Some peculiar features of antiplasticization regarding changes in Tg were probed and the effects of water were examined, both as a single component and in combination with other plasticizers to illustrate the role of intermolecular forces. The analysis has also brought to light the shortcomings of existing theories for disregarding the dual cross-over from antiplasticization to plasticization with respect to modulus variation with temperature and for not addressing failure related properties, such as yielding, crazing and fracture toughness.

## 1. Introduction and Historical Background

Plasticizers have played an important role in the utilization of polymers from their first appearance on the industrial scene to present days. Samples of artefacts at the Great London Exhibition in 1863 obtained from cellulose nitrate, under the trade name of Parkesine (later acquiring the household name of Celluloid), contained about 25–30 wt.% of camphor as an auxiliary component to make the material more easily deformed for the processing purpose [1]. The first attempt to understand the phenomenon, which became known as “plasticization”, was made in 1922 by J.B. Nichols at Cornell University (USA) [2].

A breakthrough for the use of plasticizers in polymers came with the advent of polyvinyl chloride (PVC), which made it possible to produce the very first thermoplastic elastomer [3,4]. Not only has this raised a general awareness of the potential of plasticizers as modifiers of mechanical properties of polymers but have also revealed anomalous effects at a low level of addition, consisting of an increase in tensile strength and a corresponding reduction in elongation at break (embrittlement), which later became known as "antiplasticization” to denote a change in mechanical properties occurring in the opposite direction expected from plasticization [4]. In a subsequent article Robeson and Faucher have found the same effect for the addition of Arochlor 5442 to polycarbonate and to polysulfone, and have observed a concomitant a depression of the β relaxations in the dynamic mechanical spectra of the formulations exhibiting the antiplasticization feature identified in previous work on PVC [5]. A significant insight into structural aspects of antiplasticization was the discovery that small amounts of water (a small molecule) attenuates the β relaxations of polyamide 6 (PA6) [6], which was later confirmed in a report by Dlubek et al. [7] and verified to take place concomitantly with an increase in density. However, large molecules consisting of hydroxy aromatic compounds were also found to induce antiplasticization in PA6 [8].

Later researchers have compared the effect of plasticizers at low concentrations with the similar effect brought about by physical ageing [9,10,11] in relation to changes in viscoelastic behaviour and increase in yield strength as function of temperature and strain rate [12,13,14]. These studies were continued to include crazing behaviour and fracture toughness [15].

Work on PVC was later extended to fundamental considerations, such as free volume [16], and to effects on gas diffusion [17] and to the possible role of crystallinity [18,19].

A practical benefit derived from the modulus increase brought about by antiplasticization is illustrated in the work by Delcambre et al. [20], which showed that the incorporation of small quantities of TCPP (tris 2-chloro ethyl phosphate) to polymethyl methacrylate, cast into thin films from dilute solutions in methyl ethyl ketone, increases the modulus from 3 to 5 GPa at only 5 w% addition. This resulted in a large increase in the collapse resistance of the walls of field lines produced via electrobeaming lithography.

In relation to applications for advanced technologies, Cais et al. [21] have studied the antiplasticization effects of three diphenylhydrazone derivatives on the abrasion resistance of polycarbonates as an organic photoconductor for copy printing applications. In addition to the usual increase in modulus and strength they have obtained an increase in cyclic abrasion resistance, noting also a depression of relaxations at temperatures below the glass transition.

Other examples of the benefits of antiplasticization can be derived from a review article by Ubbink [22]. The author states that at low concentrations water is observed to be a “potent” antiplasticizer in foods leading to higher mechanical rigidity (modulus) for a wide range of dehydrated systems, such as coffee beans, dehydrated apple and beef, starch matrices, including cereal-based foods, such as cornflakes and biscuits, starch hydrolysates’ and gelatin [23,24]. Farhan and Hani [25] have found that semirefined kappa-carrageenan (SRC) based edible films plasticized with either glycerol or sorbitol exhibited higher tensile strength than the corresponding unplasticized sample. Similar results were obtained by Moraru et al. [26] for mixtures of starch and polyols and also by Aguirre et al. [27]. The latter authors have also highlighted the effect of atmospheric humidity, which varied according to the type of polyol used.

Researchers have also been attracted by the inherent increase in density brought about by antiplasticization as a possible route for the enhancement of barrier properties for packaging products and protective coatings [28,29,30,31,32].

Studies on mixtures of starch and urea by Wang et al. [33] have revealed the existence of two critical concentrations, one in correspondence with a change from antiplasticization to plasticization at concentrations around 10% and another from plasticization to phase separation at around 30 w% urea.

In more recent years antiplasticization has attracted considerable attention for theoretical studies by molecular dynamics simulations, using the concept of fragility to model the cross-over from antiplasticization to plasticization [34].

In the present report we examine widely the phenomenological aspects of antiplasticization for their effect on mechanical properties and barrier characteristics for films and coatings applications, bringing out also the role of physical ageing. The deviation of free volumes from the additivity rule is examined and a model is presented for the change in morphology associated with the cross-over from antiplasticization to plasticization via the formation of plasticizer clusters, alongside the implications of data derived from studies on organic molecular glasses. Attention is given also to peculiarities and inconsistencies in the interpretation of phenomenological data.

A previous review on antiplasticization [22] addresses primarily molecular glasses on a fundamental basis and mostly in relation to structural aspects. There is, therefore, a definite need to bring together the many disjointed publications into a unifying synthesis of underlying principles. To this end we have limited the theoretical treatment to suit readers who are either not familiar with this topic or may have the expertise only in some areas. The latter would benefit from having access to the entire antiplasticization field concerned with polymers as materials for structural and functional applications.

The schematic outline (Scheme 1) of the aspects covered by this review is shown below:

The schematic outline Scheme 1 indicates that the importance of antiplasticization phenomena in polymer materials arises from the need to match properties to applications. This requires an understanding of structure/properties relationships, which is aided by considerations of polymer/plasticizer molecular interactions. These can be assessed by measuring specific volumes and through an evaluation of molecular relaxations probed by mechanical and dielectric spectroscopy.

The side frames show that there are peculiarities as well as anomalies reported in the literature concerned with antiplasticization phenomena. These cover aspects that do not fall in line with the established principles and are primarily attributed to uncertainties on the interpretation of behaviour that is contrary to what is expected from plasticization.

## 2. Structural Features and Physical Principles of Antiplasticization

### 2.1. Molecular and Morphological Considerations

#### 2.1.1. Solubility Parameters as a Criterion for Plasticizer Miscibility

Plasticizers exhibit a different level of miscibility with the host polymer and have, accordingly, been classified as “primary plasticizers” when are miscible at all concentrations and “secondary plasticizers” if they are only partially miscible [35,36].

The likely miscibility of a polymer/plasticizer combination is judged on the match of Hansen solubility parameters (δ), which consist of three components representing the different type of molecular attraction forces. This approach has its origins in the free energy of the mixing relationship with the “total” solubility parameter, which is defined as the square root of the cohesive energy density (the energy that holds molecules together within a given system).

The data in Table 1 show that the dispersive and dipolar components for tri-cresyl phosphate (TCP) are much closer than for bis-2-ethylhexyl phthalate (DOP) to the values for the monomeric units of the PVC chain. At the same time, the values for δ _H-bonding_ indicate that direct H-bonding attractions with the Cl–C–H groups are not making a large contribution to the miscibility of plasticizers in PVC. For comparison in Table 1 are included also the Hansen solubility parameters for the interactions between polyamides and several plasticizers derived by Serpe and Chaupart [8], Younker et al. [37] and He et al. [38]. The data for the latter system bring out the significant contribution from all the components of the solubility parameters, even though the miscibility for polyamides is determined primarily by the H-bonding interactions (major contributor) and secondarily by dipolar forces (minor contributor). The work of He et al. [38] has shown that the relative plasticization efficiency of three substantially different plasticizers for a newly synthesized amorphous polyamide decreased in the order water > glycerol > soybean oil.

A molecular dynamics simulation for the three systems has shown that whereas for the case of water and glycerol two types of H-bonding interactions are possible, respectively Polymer–NH····O–Plasticizer and Polymer–C=O····H–Plasticizer, only one type would operate for the interaction with soybean oil (SBO), namely Polymer–NH····O–Plasticizer. The stress/strain traces recorded from tensile strength measurements at room temperature, however, did not reveal any obvious features that could be associated with antiplasticization even at the shortest immersion time (2 days), which produced approximately 11.5 wt.% increase for water, 4.0 wt.% for glycerol and 2.3 wt.% for SBO. A large decrease in yield stress was found for all systems (>80% for water, 50% for glycerol and SBO), even at the lowest level of plasticizer absorption. These results indicate that in all cases the recorded plasticizer absorption is above the threshold concentration for the cross-over from antiplasticization to plasticization with respect to mechanical properties measured at ambient temperature (discussed later).

#### 2.1.2. Polymer/Plasticizer Interactions by Fourier Transform Infrared Spectroscopy (FTIR)

Paris and Coupry [39] have identified the occurrence of strong H-bonds in cellulose nitrate plasticized with camphor. The C=O stretching mode at 1741 cm^−1^ in the camphor spectrum appeared at 1730 cm^−1^ in mixtures with cellulose nitrate. On the other hand, Benazzouz et al. [40] found that the shift in C=O absorption peak in cellulose acetate (CA) from 1736 to 1751 cm^−1^, resulting from the addition of dimethyl phthalate (DMP) plasticizer, is not attributable to H-bonding between the C=O groups in DMP and the residual OH groups in CA.

In an Fourier transform infrared spectroscopy (FTIR) examination of the H-bond interactions between different types of sulphonamide plasticisers in several polyamides 12, De Groote et al. [41] were able to determine the relative fraction of NH groups in the polyamides involved in the interaction with the sulphonyl groups of the plasticizer by monitoring the upward shift of the absorption peak at 3322 cm^−1^, as well as the breadth and details of the absorption band through a decomposition of the constituent peaks.

The complexity of H-bonding interactions in plasticised polymers is revealed in the work by Wang et al. [33] on mixtures of starch and urea, where it is shown that several groups can participate in H-bond formation. These authors noted that the absorption peak at 989.5 cm^−1^ for the C O H bending vibration in the starch molecules was shifted to 991.4 cm^−1^ in the presence of urea, while the absorption peak at 927.8 cm^−1^ attributed to the skeleton mode vibrations of the C O C, changed to 929.7 cm^−1^ at 5% urea and increased to reach a plateau value of 933.5 cm^−1^ at concentrations greater than 15 wt.% urea, which corresponds to the solubility limit for the system.

Musto et al. [42] have used the concept of dual nature of water in a post-cured tetrafunctional epoxy resin using FTIR in the mid-infrared (4000–400 cm^−1^) and in a near-infrared (8000–4000 cm^−1^). In this study the spectrum of the absorbed water molecules was isolated in both frequency ranges and assignments of the various components of the spectrum were made. From these it was possible to confirm that the absorbed water was present both as a mobile variety residing within “microvoids” (free water) and as molecular associations with the polymeric network through hydrogen bonding interactions (bound water).

#### 2.1.3. Plasticizer Clustering

Since the solubility of the plasticizer relies on interactions with constituent groups along the polymer chains, at low concentrations the plasticizer molecules can be envisaged as being “bound” to the polymer chains and restrict the rotations of short segments (β relaxations), giving rise to an antiplasticization behaviour. At higher concentrations not all the plasticizer content can be accommodated as “bound” molecules along the polymer chains and, therefore, the excess plasticizer will start to segregate into small “clusters” of molecules in a dynamic state of association/dissociation with the polymer chains, giving rise to the initial stage of the cross-over to a plasticization behaviour. It should be noted that the presence of segregated plasticizer “microclusters” has also been identified by Bergquist et al. [43] for mixtures of polycarbonate and tris(2-ethyl hexyl) phosphate. The dimensions were estimated to be in the region of 1.2 nm, consisting of two adjacent plasticizer molecules. It is envisaged that the dispersed clusters will grow in number with increasing plasticizer concentration until they become interconnected and reach the “full plasticization”. This mechanism for the conversion to plasticization is thermodynamically more favourable than clusters coalescing into larger domains owing to the higher “strength” of polymer–plasticizer interactions over cohesive plasticizer–plasticizer attractions.

### 2.2. Free Volumes in Relation to Plasticization and Antiplasticization

#### 2.2.1. The Additivity Rule, Free Volume and “Holes”

The physical state of a glass subsumes that the specific volume has two components, respectively a volume “occupied” by the atomic constituents of molecular chain and a “free” volume between polymer chains. The concept of fractional free volume (FFV) has often been used as a quantitative parameter for the state of molecular packing of glasses (i.e., FFV = *V*_free_/*V*_total_). The change in free volumes for a non-miscible mixture of the polymer (1) and plasticizer (2) can be estimated using the additivity rule, so that the specific volume, ψ_sp_, (reciprocal of density) can be calculated from the equation:
ψ_sp/12_ = ω_1_·ψ_sp/1_ + ω_2_ ψ_sp/2_(1)
where ω_1_ and ω_2_ are the respective weight fractions for the polymer and plasticizer.

This equation can be extended to include the temperature dependence of individual specific volume of the specific components [44]. For temperatures within the glassy state of the mixture the modified expression becomes:(2)$$\psi_{\mathrm{sp}/12}\;(T)=\omega_{1}\cdot \psi_{\mathrm{sp}/1}(T)+\omega_{2}\;\psi_{\mathrm{sp}/2}(T)+\omega_{1}\;(\frac{d\left(\psi_{\mathrm{sp}/1L}\right)}{d\;T}-\frac{d\left(\psi_{\mathrm{sp}/1G}\right)}{d\;T})\;({\mathrm{Tg}}_{12}-T)$$
where Tg_12_ is the glass transition temperature of the mixtures and the subscripts L and G stand for the liquid (rubbery) state and glassy state. An alternative way to express the temperature effect has been used by Zhang et al. [45] in terms of the volumetric relaxation rate.

Using the locally correlated lattice (LCL) model Roland et al. [46] have calculated the total free volume and derived a predictive relationship with the glass transition temperature. Accordingly, the authors have demonstrated that a melt transform into a glass by cooling upon reaching a “boundary” value for the minimum total free volume, which decreases approximately linearly with changes in temperature. The results obtained by the LCL model are consistent with the theory that the difference in entropy between the melt state and the ensuing solid state vanishes as the glass transition is approached [34].

Antiplasticization is manifested at low plasticizer concentrations as a minimum with respect to the variation of a specific volume, which has been associated with a negative deviation from the additivity rule (Equation (1). This results from a combination of two related events: reduction in free spaces between polymer chains [47] and enhanced intermolecular attractions with the polymer chains.

Studies carried out by positron annihilation spectroscopy (PALS) have estimated the radius of the so-called “holes” (related to the free volume) in glassy polymers to be in the region of 0.26–0.27 nm, while the volume is estimated to be in the region of 5.0–8.0 nm^3^ [7,41,48,49]. To place these dimensions in perspective the radius of the “hole” is approximately equal to the radius of the sphere occluding the structure of urea [50] and twice that of a water molecule. It is also worth noting that Caldwell and Jackson in their pioneering work in 1967 [51] had already deduced that the molecular dimensions of antiplasticizers for polycarbonate should be less than 0.55 nm in at least 65% of the length of the molecule, which corresponds approximately to the size of a PALS “hole”.

In studies carried out on mixtures of different grades of polystyrene with mineral oil Anderson et al. [52] have deduced that antiplasticization occurs when the average diameter of the mineral oil domains is less than the average size of the free volume voids. The hole size for the polymer was estimated to be 0.57 nm, while the size of the mineral oil domains at the maximum concentration (6 wt.%) for antiplasticization was reckoned to be in the region of 0.2 nm.

#### 2.2.2. Deviations from the Additivity Rule

In early studies on antiplasticization the deviation from the additivity rule was quantified thorough the incorporation of an interaction parameter (*k*) in Equation (1), which becomes [53,54]:ψ_sp/12_ = ω_1_ ψ_sp/1_ + ω_2_ ψ_sp/2_ + *k* ω_1_ ω_2_(3)
where *k* is obtainable from the experimental data.

Increasing the plasticizer concentration above the minimum in the relationship with the specific volume results in a further increase and then reaches a certain “threshold” value, *ω_1(T)_*, when the specific volume of the mixture becomes equal to that of the pristine polymer. The specific volume will subsequently rise further into a highly plasticized state of the polymer mixture.

A useful approach to the deviation of plasticized systems from the additivity rule arising from molecular packing of the polymer/plasticizer mixture [47] is to consider the decrement of free volume Δ(FV), which can be estimated using the following equation:(4)$$\%\;\Delta (\mathrm{FV})=-(\frac{\psi_{\mathrm{sp}(*)}-\psi_{\mathrm{sp}(\omega )}}{\psi_{\mathrm{sp}(*)}})\;\times \;100$$
where ψ_sp(ω)_ is the measured specific volume of the polymer/plasticizer mixture at concentration ω, and ψ_sp_${}_{\left(*\right)}$ is the theoretical specific volume of the mixture at the same plasticizer concentration.

We have calculated the values of % Δ(FV) for the data presented in Figure 3 (discussed later) using estimates for ψ_sp_${}_{\left(*\right)}$, obtained according to Equation (1). The obtained values, together with various related parameters, are shown in Table 2.

The values for the free-volume decrement factor, % Δ(FV), indicate that the deviation from the additivity rule continues well within the plasticization regime. This suggests that the amount of bound water may continue to increase during the formation of clusters, possibly at a much lower rate than that within the antiplasticization range of conditions. The data in Table 2 indicate that there is a maximum % Δ(FV) factor at around 30% water content, which arises from the inevitable convergence of the trend line to zero as the water fraction approaches the value of 1.

Maeda and Paul [55] have used the concept of “excess free volume, Δ(Ve)” for the discrepancy between calculated and measured values of specific volume, which corresponds to a non-normalized deviation, i.e.,
(5)$$\Delta (\mathrm{Ve})=-\left(\psi_{*\mathrm{sp}}-\psi_{\mathrm{sp}(\omega )}\right)$$

We have performed the same calculations for mixtures of poly(2,6-dimethyl-1,4-phenylene oxide (PPO) with di-octyl phthalate (DOP) and tricresyl phosphate (TCP), where the plasticizer is much bulkier than water and the polymer/plasticizer interactions are predominantly dipolar in nature, rather than H-bonding type. The values were obtained using data reported by Maeda and Paul [55] and are compared in Table 3 with the values estimated from Equation (4).

The data in Table 3 revealed a similarity in the effect of plasticizer concentration for both PPO systems, irrespective of the factor used to determine the deviation of free volume from the additivity rule. These show also that the maximum for both Δ(Ve) and Δ(FV) occurred at approximately 10 wt.% concentration for the starch/water system, while for plasticized PPO systems the maximum spanned over the range 20–30 wt.% plasticizer content. This implies that smaller plasticizer molecules and stronger interactions produce a maximum in the deviation at lower concentrations.

In an atomic simulation of changes in free volumes of a “model” amorphous polyamide 6.6 (PA 6.6) plasticized with different amounts of water, Goudeau et al. [56] have used the concept of “additional free volume”, Φ, defined as
(6)$$\Phi =\frac{-\left[\psi_{*\mathrm{sp}}-\psi_{\mathrm{sp}(\omega )}\right]}{\psi_{\mathrm{sp}\left(\mathrm{polymer}\right)}}$$
which corresponds to the value of Δ(Ve) normalized with respect to the polymer. For comparison purposes we recalculated the values normalized with respect to ψ*_sp_ from the simulation data at 280 K for water contents of 5 and 10 w%, corresponding to the free volume decrement factor of Equation (4). The values obtained were −1.59% at 5 wt.% water and −3.95% for the system containing 10 wt.% water, which are intermediate between the PPO systems and the starch/water mixtures. These differences have provided a basis to reveal the stronger molecular attractions in PA 6.6/water mixtures relative to the PPO systems and the greater packing efficiency for the starch/water mixture.

Another example of deviation from the additivity rule for property prediction is to be found with respect to the water absorption of mixtures of Kollidon VA64 (random copolymer of vinyl pyrrolidone and vinyl acetate) and clotrimazole [57]. The mixtures were found to be miscible at all concentrations. The data relevant to this discussion are shown in Figure 1 as plots at two different levels of environment relative humidity (RH) for the “water vapour absorption” (left) and the “hardness” measured on the films (right) as a function of the clotrimazole weight fraction. The plots on the left show that while the water absorption of the mixtures (calculated by the additivity rule on the similar basis as that for Equation (1) decreased linearly with increasing clotrimazole content, the measured values were consistently lower. The two plots in the inset show that the difference between calculated and measured values (deviation) exhibited a maximum at the estimated deviation at plasticizer concentrations in the region of 50 wt.%.

The plots on the right for hardness measured on the same mixtures show a corresponding antiplasticization effect with the maximum displayed to higher concentrations. Since both properties were density dependent, the discrepancy with respect to the maximum indicates that the two were not linearly related, as it was also observed in the discussion of the data in Table 3.

#### 2.2.3. Comparison of Antiplasticization with Physical Ageing

A better understanding of antiplasticization can be obtained by making a comparison with the densification arising from physical ageing of glassy polymers, illustrated in Figure 2, as both resulted in an embrittlement behaviour alongside an increase in modulus and strength.

The representative lines shown in Figure 2a indicate that densification by physical ageing arises purely from the reduction in the free volumes of polymer chains from the “juvenile” (unaged) state until the glass reaches a “mature” (minimum free energy) state, characterized by its equilibrium Tg (also known as Tα). The trace for a “mature” glass indicates that the change in the specific volume with a temperature below the Tg of the polymer was expected to take place at the same rate as that of the volume occupied by the polymer chains in relaxations-free state, while the constant gap was to allow for a likely temperature independent compressibility.

The descriptive lines for Figure 2a were derived from the changes in modulus with temperature within the glassy state outlined in the work of Mascia et al. [9,12], shown in the inset, and also from a report by Lee [58], as well as from an article by Lee and McGarry [59] on atactic polystyrene. The graphs in Figure 2a illustrate the connection between changes in the specific volume (ψ_sp_) and shear modulus (G) with temperature for a glassy polymer undergoing physical ageing through the relationship between relaxation time and free volume in Equation (11) (i.e., d(*log*G)/dT is proportional to −dψ_sp_/dT_)._ One notes that the rate of change of both functions decreases with ageing in advancing towards a “mature” (stable) state and brings about an upward shift in the glass transition temperature. The trendlines for the specific volume bear a similarity with the experimental data reported in [59], showing a reduction in the gradient of the change in the specific volume at temperatures below 50 °C after physical ageing. In this latter case the trendlines for the aged and unaged samples allude to a likely intersection at lower temperatures, possibly about −50 °C in correspondence to a more realistic temperature for the β transition of polystyrene [60], in concordance with the values for other glassy polymers. It should also be noted that the graphs representing the variation of specific volume with temperature in [59] indicate that the Tg of the sample decreased with increasing ageing time. Since this is physically impossible the discrepancy may have to be attributed to experimental errors in the dilatometry measurements, bearing in mind that an increase in Tg has also been observed in related experimental studies by Yavari et al. [61].

The behaviour depicted in Figure 2b, on the other hand, is derived from the singular log-linear relationship between modulus and specific volume found by Maeda and Paul [62], which is consistent with the numerical simulation analysis by Puosi and Leporini [63] and the experimental data of Zhang et al. [64]. It should be noted, however, that behaviour at temperatures below the β transition has not been addressed in the literature on antiplasticization and, therefore, the traces shown are somewhat speculative. In any case, a more accurate description of the variation of the specific volume would require the trend line for the antiplasticized system (1) to intersect the one for the pristine polymer (system 0) at a temperature between Tα and Tβ, resulting from a reduction in thermal expansion coefficient [65].

A glassy polymer containing an amount of plasticizer within the antiplasticization regime will be susceptible to accelerated physical ageing as a new system with a lower Tg [66], arising from a delay in the “freezing” of the molecular motions during vitrification, which would accordingly depress the β relaxations and produce a denser glass [59].

In the forgoing discussion the antiplasticization threshold was identified as the plasticizer concentration, ω_1(T)_, for which the specific volume of the binary system becomes equal to that of the pristine polymer at any temperature below the Tg. This interpretation is illustrated in Figure 3 using data obtained by from measurements of the density of mixtures of amorphous starch and water [67]. The antiplasticization behaviour represented in Figure 3 is concordant with the model proposed by Wang et al. [33], as discussed in the preceding section with respect mixtures of starch and urea. Moreover, the above interpretation for the changes in free volumes at different water (plasticizer) content is supported also by complementary data in part 2 of the paper by Benczedi et al. [68] in relation to the calculated cohesive energy. The findings are consistent with the model of plasticizer molecules being “physically bound” to the polymer chains, which results in an increase in the “effective” molar volume of the structural units [69], thereby reducing the size of molecular holes [70].

In examining the plotted data in Figure 3 one notes that the points for the increase in the specific volume after the minimum can be fitted to two intersecting straight lines. The lower gradient for the first line (after the minimum) can tentatively be attributed to the formation of “dispersed clusters” of water (plasticizer) molecules. A value of 2.75 wt.% can be identified as the amount of the bound water before the onset of cluster formation, whereas a value of 10.5 wt.% for the threshold condition for the cross-over from antiplasticization to plasticization. It must be emphasized, however, that there is no confirmatory experimental evidence for the general validity of the two-gradient feature displayed in Figure 3, as well as for the formation of interconnected clusters. The formation of coarse domains, on the other hand, has been shown to take place from studies on the interactions between diethyl phthalate and cellulose acetate (CA) [36], which was found to exhibit two Tg values at high plasticizer concentration.

#### 2.2.4. Molecular Weight Considerations

In studies carried out to examine the effect of the molecular weight (MW) of polystyrene (PS) in miscible mixtures with mineral oil, Anderson et al. [52] found that antiplasticization occurred only when the MW was relatively low (40,000 Da) at oil concentration in the region of 6 w%. Such a behaviour was attributed to the presence of a large number of chains ends, which are the main sites for the formation of “holes” as the major contributors of free volumes and were estimated to be about 80 times greater in number than the grade with MW = 270,000 Da. Moreover, the authors have proposed that phase separation takes place within the plasticization range of concentrations due to the stronger interactions for plasticizer–plasticizer than for polymer–plasticizer. This is due to the higher (total) solubility parameter of the oil relative to polystyrene, which is in concordance with the model proposed earlier for conditions that are not favourable for the formation of interconnected clusters. A similar behaviour is exhibited by the starch/water mixtures depicted in Figure 3, which provides additional evidence for the formation of plasticizer clusters. In any case, it should be borne in mind that the observations and measurements were made at room temperature and that the threshold conditions for the antiplasticization/plasticization transition are temperature dependant (as indicated earlier). A similar argument for plasticizers used for toughening of polylactic acid (PLA) [71]. The authors did not detect antiplasticization behaviour from studies of the relaxation spectra of mixtures of PLA and acetyl tributylcitrate (ATBC) even at concentrations as low as 2.5%. Since the molecular weight of the amorphous PLA used in this study (reported to be in the region of 90,000 Da) was considerably higher than the “threshold” value required for the onset of antiplasticization in the case of PS [52], it can be inferred that the lack of a sufficient number of chain-end groups as a criterion applies also to PLA/ATBC mixtures. This may provide the basis for a general criterion for antiplasticization, which would include molecular-weight considerations alongside other factors such as polymer–plasticizer interactions and dimensions of the plasticizer molecule.

Within the context of the above analysis it is also worth noting that, in studies carried out on physical ageing of PLA, Pan et al. [72] have correlated the highly impaired ductility resulting from physical aging to the rearrangement of polymer chains from disordered to more ordered morphology. By analogy the propensity of glassy polymers to undergo antiplasticization could be related to the ability to produce a certain degree of structural order, which would be easier to achieve at lower molecular weights.

### 2.3. Molecular Dynamics of Antiplasticization

#### 2.3.1. Relaxations Interpretation from Mechanical and Dielectric Spectra

The response of polymers to mechanical stresses is described by viscoelasticity theory, whereby when an external excitation is imposed as sinusoidal wave transmitted through the bulk the corresponding strain is out-of-phase by an angle *δ* (the loss angle. The proportionality coefficient (modulus E = stress/strain) is expressed as a complex entity consisting of two components, one in-phase (storage modulus) and another out-of-phase (loss modulus), i.e.,
E* = E′ + *i* E″, tan *δ*_(mech)_ = E″/E′(7)
where *i* is the complex number ($\sqrt{-1}$).

Similarly, the relationship between an applied alternating electrical stress and the resultant charge density is expressed by the permittivity in complex notation (ε* = charge density/electric stress), i.e.,
ε* = ε′ − *i* ε″, tan *δ*_(diel)_ = ε″/ε′(8)

The out-of-phase (loss) terms of the expressions in Equations (7) and (8) are both frequency and temperature dependant and will go through a maximum at a thermal transition. Although the loss component is the part of the spectrum that represents the energetic events leading to the thermal transitions, tan *δ* plots as function of temperature (at low frequency, usually 1 Hz) are often used to analyse the data. The temperature/frequency relationship is expressed in terms of activation energy of the thermal transition, using the Arrhenius rate equation as the variation of the related relaxation time, *τ*, with temperature (see Equation (9), later).

Glassy polymers are characterized by three transitions, respectively α transition, corresponding to glass transition (Tg) at the upper temperature end of the spectrum, an intermediate β transition (associated with JG-β relaxations in dielectric spectroscopy) and a γ transition at the lower temperature end of the spectrum. The α transition has the highest activation energy and the γ transition the lowest.

The changes in mechanical spectra brought about by physical ageing and antiplasticization are illustrated in Figure 4a,b. These show that physical ageing is manifested as a depression of β relaxations spanning to the lower end of α transition regions, without significantly affecting the peak temperature of the two transitions. Although antiplasticization reduces the overall relaxation time for α transition and lowers the glass transition temperature, the effect on the β transition is manifested mainly as a depression of the relaxation spectrum (increasing the inherent relaxation time), which is accompanied only by small changes in the temperature of the corresponding tan *δ* peak.

However, a clear shift in the β relaxation peak is not always discernible and may even appear to remain invariant or shift slightly upward on the temperature scale [10,73]. For cellulose acetate containing different amounts of diethyl phthalate (DEP) [36,74] the small downward shift of the β transition peak observed at low concentrations did not increase significantly at high DEP concentrations. Similar results were obtained by Seymur et al. [75] while the work of Lourdin et al. [76] provides further confirmation of the depression of β relaxations observed at low concentrations of glycerol in starch systems containing 13% water. Although the position of the peak of the weak β relaxations could emerge as an artefact introduced by the normalization of the loss component relatively to the “real part”, dielectric spectroscopy data obtained by Psurek et al. [77], on the effect of small quantities of Arochlor 1260 in polycarbonate, have revealed a depression of the β transition with a slight downwards shift in the broad peak, even when the actual loss component, ε”, (instead of tan *δ*) was plotted against temperature.

From a comparison of the relaxation features shown in Figure 4a,b, it can be deduced that the depression of β transitions brought about by antiplasticization might be enhanced through an ensuing more advanced physical-ageing state of the glass under the same thermal history conditions. Accordingly, Cangialosi et al. [78] have indicated that the depression of β relaxations by the two connected events can be used as a measure of the densification efficiency of the system.

The activation energies (Ea) for relaxations are obtained using the Arrhenius equation for the relationship between the imposed oscillation frequency (*f*) and temperature (T) at which the tan *δ* relaxation peak occurs, i.e.,
*f* = A exp(−Ea/RT)(9)
where R is the gas constant (8.314 J/mol/K) and A is a material constant. This principle is based on the hypothesis that a high proportion of the polymer chains react isochronally with the applied frequency.

In relation to the effects of plasticizer in mixtures with polyvinyl chloride Elicequi et al. [79] have observed that while the activation energy for the α transition decreased there was no noticeable change for the β transition. The authors have related this behaviour to the shallow and broad nature of the β relaxation band. An unusual feature emerged from recent work by Maeda et al. [80] on antiplasticized mixtures of PC, using rigid chain plasticizing molecules (length 1.1–1.5 nm). The data have also revealed the emergence of a β relaxation shoulder at about 80 °C, alongside a reduction the α relaxation peak, which was attributed to the concomitant reduction in the γ relaxation peak at around −100 °C. The activation energy for the γ relaxations was found to be higher in the presence of plasticizing species, while the values for β relaxation were found to increase with the molecular length of the plasticizer.

#### 2.3.2. Melt Fragility as a Parameter for Antiplasticization

The concept of fragility is widely used in theoretical studies on vitrification of glasses from a liquid, or “melt” state. It represents a parameter that characterizes the magnitude of the deviation of a physical rate-controlled property, such as viscosity, from the Arrhenius equation at temperatures very close to the Tg (α transition). Angell has used the terms *fragile, intermediate* and *strong* as a designation for the glass forming characteristics of the material [81]. A strong melt was defined as one that obeys the Arrhenius solidification rate law for the change in viscosity (or related relaxation time), which implies that the glass retains its physical state at all temperatures. A fragile melt, on the other hand, displays a very large deviation from the continuity of a physical state through vitrification, which alters the behaviour from a liquid (viscosity controlled) to a solid (modulus controlled) on cooling. The variation of relaxation time (τ) that characterizes the physical state with temperature (T) upon reaching the glass transition temperature is usually described by an empirical equation proposed by Vogel, Fulcher and Tammann, which became known as the VFT equation [82], i.e.,
(10)$$\tau =\tau_{\mathrm{VFT}}\cdot \exp\;(\frac{B}{T-\mathrm{To}})$$
where τ_VFT_ is the pre-exponential factor and B is a material constant, often replaced by D · To, where D is referred to as a “stiffness” or “fragility” parameter that quantifies the “strength” of the dependence of τ on temperature, while To is usually known as the “Vogel divergence temperature”, whereby T_o_ < Tg and T > Tg. To is sometime referred to as the “ideal glass transition temperature” in so far it represents the temperature at which the entropy extrapolates to zero. The parameters of Equation (10) are related to free volume (*Vf)* by the Doolittle equation through the expression [83]
(11)$$\tau =\tau_{o}\exp\;(\frac{V}{Vf})$$
which alludes to the existence of free volumes as a requirement for molecular relaxations.

This relationship provides the basis for a general description of glass formation through the definition of a characteristic melt fragility factor (*m_p_*), i.e.,
(12)$$m_{p}=(\frac{d\log\;\tau \alpha }{d\left(\frac{\mathrm{Tg}}{T}\right)})P\;\mathrm{for}\;T=\mathrm{Tg}\;(\mathrm{under}\;\mathrm{isobaric},\;P,\;\mathrm{conditions})$$
where τ *α* is the relaxation time for the melt as it approaches vitrification conditions.

The theoretical work by Stukalin et al. [84], based on the relationship between structural relaxation times and the configurational entropy density, predicts that antiplasticization reduces the fragility of glass formation when the plasticizer forms strong interactions with the polymer matrix, which is brought about by corresponding increase in the relaxation time for the β transition. The model used by these authors relates the effects of cohesive energy, chain length and stiffness on the antiplasticization and plasticization behaviour of miscible additives.

## 3. Implications of Antiplasticization for Applications

### 3.1. Mechanical Properties

#### 3.1.1. Modulus Enhancement through Antiplasticization

##### Experimental Data

The diagrams in Figure 5 show the variation in complex modulus at an arbitrary low frequency (1 Hz) of a glassy polymer with temperature obtained by dynamic mechanical spectroscopy (DMA) methods. The curves in Figure 5a represent the behaviour of samples subjected to physical ageing, while those in Figure 5b corresponding to systems containing plasticizer amounts below and above the antiplasticization threshold. Note the trend lines in Figure 5b are adapted from original data reported for plasticization of PVC with tricresyl phosphate [10,12,15,77]. The cause of the dual cross-over feature for the plasticization/antiplasticization phenomenon has yet to be addressed in the literature either through theoretical modeling or experimental verifications from specific volume measurements. It is possible that this behaviour is related to morphological changes induced by the plasticizer propensity to phase separation through internal associations driven by crystallization or even vitrification.

These diagrams show that there were two threshold temperatures for the plasticization/antiplasticization cross-over. An inset has been added to Figure 5 as an experimental confirmation of the indicated trend, using an extract of data reported by Psurek et al. [77] for studies on antiplasticization of polycarbonate (PC) in mixtures with Arochlor 1260. Here **a** denotes the pristine PC sample and the letters **b**, **c**, **d** and **e** stand for mixtures with an increasing concentration of plasticizer. This feature of the antiplasticization behaviour of PC is also confirmed by the work of Miyagawa et al. [65] in mixtures with *p*- terphenyl as plasticizer.

##### Modelling Prediction of Elastic Constants

Model studies have been carried out by Riggleman et al. [85] to simulate the antiplasticization/plasticization cross-over using the concept of variation of fragility and the relationship between relaxation time and configurational entropy. The model used predicts the variation of Young’s modulus, shear modulus and Poisson ratio with respect to the temperature and establishes the cross-over conditions. The model employed in this work consists of a bead-spring polymer chain and smaller, spherical solvent (plasticizer) molecules. The polymer molecules were modelled as 32-segment chains, where each segment was connected via a stiff harmonic potential. Two sets of non-equilibration molecular dynamics calculations were performed to obtain the bulk mechanical properties. The authors have used uniaxial tension deformations at an arbitrary constant strain rate to obtain the Young’s modulus.

The shortcomings in the predictions of the model used by these authors are twofold: (a) The model does not allow for the possibility of a reversion to plasticization at some lower temperatures, as indicated by the experimental data presented in the inset of Figure 5b and (b) the model does not address failure related properties, such as yielding, crazing and brittle fractures. A revealing feature of the model, however, is the prediction of a lower Poisson ratio within the antiplasticization temperature region and an inversion of behaviour at higher temperature within the plasticization regime. Although not directly confirmed experimentally, the verification is implicit in the measured increase in the volumetric strain resulting from uniaxial tensile stresses for tests on an antiplasticized epoxy-resin material reported by Garton et al. [86] (Section 3.1.2). A similar inference can be derived from the reduced crazing strain values for an antiplasticized PVC material [15], which can be associated with the hydrostatic stress component of the applied tensile stress (see Section 3.1.2, relationship between *ε_1(craz)_* and *I* in Equation (17).

#### 3.1.2. Evaluation of Data from Strength Measurements

Mascia et al. [12] have demonstrated that the antiplasticization double cross-over is displayed also with respect to yield strength, using conventional tensile tests and plane–strain compression experiments carried out at different strain rates over a wide range of temperatures. Although the maximum strain rate and minimum temperature used in tensile tests were limited by the occurrence of brittle fractures, the data in Figure 6 reveal not only the strain rate sensitivity of the antiplasticization/plasticization cross-over but also the effect of dilatational stress components in tensile tests. In this latter case a trend-line extrapolation suggests that the upper cross-over was displaced to a lower temperature when the strain rate was reduced.

The two cross-over temperatures for the yield strength in compression under plane strain conditions, on the other hand, do not seem to be greatly affected by the strain rate. These observations indicate that the thresholds conditions for the cross-over from antiplasticization to plasticization with respect to mechanical properties depend also on volumetric changes, i.e., dilatation (tension) or contraction (compression), resulting from the mean stress component of the applied stresses. This deduction is supported by the data reported by Garton et al. [86] for the tensile strength of a cured epoxy resin containing antiplasticizer EPPHAA (an adduct of the reaction between epoxyphenylpropane and 4-hydroxyacetanilide). The authors have observed a steeper rise in the volumetric dilatation with increasing strain in tensile tests for samples containing the antiplasticizer. Similar results were obtained by Mikus et al. [87] from tensile tests carried out on mixtures of starch and glycerol, sorbitol and mannitol. From measurements of the strain in the three perpendicular directions, using a video system, the latter authors were able to separate the elastic deformation from plastic deformation and to estimate the variation of volumetric expansion as function of the axial strain, i.e.,
ε_v_ = ln (*V*/*V*_o_) = ε_1_ + ε_2_ + ε_3_(13)
where ε_v_ = volumetric strain, *V* = volume at strain level considered, *V*_o_ = volume in the unstrained state and ε_1_, ε_2_ and ε_3_ are the respective strains in the principal directions.

Accordingly, the volumetric strain is considered to include both an elastic (ε_el_) and an inelastic (ε_inel_) component, i.e.,
ε_inel_ = ε_v_ − ε_el_(14)
where ε_el_ is calculated from the expression
(15)$$\varepsilon_{\mathrm{el}}=\frac{\left(1-2\;\vartheta \right)\sigma }{E}\;\;$$
with σ = axial tensile stress, *E* = Young’s modulus and ϑ = Poisson’s ratio.

These authors found that the volumetric expansion resulting from the applied axial stress decreases with increasing concentration of the plasticizer and is reduced by physical ageing. This can be attributed to an increase in bulk modulus and can, therefore, taken as a manifestation of antiplasticization behaviour.

#### 3.1.3. Fracture Toughness and Related Phenomena

Mascia et al. [15] have evaluated the embrittlement effect of antiplasticization of glassy linear polymers from measurements of the variations of the “critical stress intensity factor”, Kc, and the “critical strain for crazing”, ε_(craz)_, both as function of temperature.

The concept of critical stress intensity factor derives from fracture mechanics as a stress related property that characterizes the strength of the material in the presence of a sharp notch (“crack”), is normalised using the expression:(16)Kc=Yσ √a  
where σ is the tensile stress at fracture, Y is a geometry related factor and a is the initial crack length.

Crazing is a unique feature of thermoplastic glassy polymers, which occurs ahead of a propagating crack and is a precursor to brittle fractures. A critical crazing strain criterion was introduced by Oxborough and Bowden [88] as a modified version of the Sternstein and Myers criterion based on the tensile stress difference between two principal axes [89].

The critical strain criterion is expressed in terms of maximum tensile strain at which crazes appear, i.e.,
(17)$${\varepsilon_{1}}_{\left(\mathrm{craz}\right)}={\left(\sigma_{1}\right)}_{\mathrm{craz}}E=\sigma_{1}/E-\vartheta \;(\sigma_{2}+\sigma_{3})/E=A(T)+\frac{B\left(T\right)}{I}$$
where *E* = Young’s modulus, ϑ = Poisson’s ratio and *σ_1_*, *σ_2_* and *σ_3_* are tensile stresses along the three principal axes. I = *σ*_1_*+ σ*_2_
*+ σ*_3_ (*I* = 3 *σ*_(hydrostatic)_), while *A(T)* and *B(T)* are temperature and environment dependant materials parameters.

In this work the critical crazing strain was determined under plane stress conditions by carrying out experiments at various temperatures on thin-and-wide rectangular specimens, bent over mandrels of different diameters and recording visually the formation of crazes. The axial (outer skin) strain at which crazes were observed was estimated according to the equation below [90]
(18)$${\varepsilon_{1}}_{\left(\mathrm{craz}\right)}=\frac{1}{2\left(\frac{R}{h}\right)+1}\;\;$$
where R = radius of the mandrel and h = thickness of the sample.

To quantify the magnitude of the antiplasticization effect it is useful to use the concept of “Antiplasticization property ratio” (*APR*), as a normalized “property parameter” of the mixture relative to the pure polymer, i.e.,
(19)$$\mathrm{APR}=\frac{Property\;of\;mixture\;}{Property\;of\;polymer}\;$$

Accordingly, *APR* values greater than 1 are obtained when antiplasticization brings about increment in the property considered.

In Figure 7 there are reported plots of the *APR* values for the critical crazing strain and stress intensity factor against temperature from the work of Mascia et al. [15] on PVC containing 8.5 wt.% TCP. The data show that the *APR* values for both ε_(craz)_ and Kc are less than 1 over a temperature range between the α and β transition.

These diagrams demonstrate that the two threshold-temperatures concept for the crossover from antiplasticization to plasticization applied also to fracture behaviour, respectively one at low temperatures (T_(A/P)1_) close to the β transition and one at a temperature below the Tg of the polymer (T_(A/P)2_). Moreover, the plots in Figure 8 bring out the advantage of the *APR* factor as a quantitative parameter for antiplasticization, insofar these show that not only the obtained values are lower for crazing but also that the cross-over to plasticization (T_(A/P)1_ and T_(A/P)2_) occurs over a narrower temperature range (i.e., −50 to +40 °C for crazing and −80 to +60 °C for fracture toughness). The data are in good agreement with the known relationship between crazing and fracture and provide a clear indication that the embrittlement of thermoplastic glassy polymers through antiplasticization arises from the enhanced susceptibility to crazing. This observation can also be connected to the reduced dilatability of an antiplasticized system identified by Mikus et al. [87], insofar as crazing originates from the hydrostatic component of the applied stress (see relationship between *ε_1_*_(craz)_ and *I* in Equation (17). It is worth noting that a similar “double trough” trendline has been observed also for the impact fracture toughness behaviour of polymethyl methacrylate (PMMA) [91].

### 3.2. Permeation and Diffusion Related Properties

The relationship between the permeability, *P*, of a gas through a film and the free volume of the polymer is described using the concept of fractional free volume (*FFV*), which corresponds to the fraction of the total volume not occupied by polymer molecules, i.e.,
(20)$$P=Po\;\exp\left(-\frac{B}{FFV}\right)$$
where *Po* and *B* are adjustable constants depending only on temperature and penetrant type.

A similar expression is used to describe the diffusion coefficient as a function of fractional free volume, known as the Doolittle expression [92]: (21)$$D=Do\;\exp\left(-\frac{A}{FFV}\right)$$
where, *D_0_* and *A* are constants, which are determined by the polymer-penetrant system.

This equation has been modified by Cohen and Turnbull [93] by considering the probability of a diffusing molecule finding a hole large enough within the glass, i.e.,
(22)$$D=Do\;\exp\;(\;\frac{-\gamma V*}{Vf}\;)$$
where γ = numerical factor that accounts for the sharing of free volumes by neighbouring molecules, V* = minimum hole volume size required and Vf = average free volume for spherical molecules in the liquid state.

Equations (21) and (22) have been used by several authors [94,95] to verify the effect of *FFV* on gas permeability in glassy polymers.

#### 3.2.1. Gas Permeation Barrier and Membranes

The role of antiplasticizers in polymers with respect to the permeation of gases was first identified in the pioneering work of Robeson [96] who reported an almost 3-fold reduction in the CO_2_ permeation through polysulfone films with the addition of 10 w% 4-4’ dichlorodiphenyl sulfone. A more dramatic decrease in permeability was reported by Maeda and Paul [97], showing an almost 30-fold reduction with the incorporation of 30 w% of *N*-phenyl-2-naphtylamine for the same type of polysulfone and a 10-fold reduction for polyphenylene oxide containing 30 w% tricresyl phosphate.

Sometime later Guo [98,99] studied the effects of physical ageing and antiplasticization on the water transport properties of glassy cellulose acetate film-coated tablets. Physical ageing was found to have an additional effect on antiplasticization, which prompted the authors to propose a model that considers the free volume of glassy polymers to consist of two independent parts, one of which is affected by annealing (physical ageing) and the other by antiplasticization. It was suggested that at low concentrations not all the free volumes are occupied by the plasticizer and, therefore, they are affected independently of each other. Horn and Paul [100] and Xia et al. [101] have reiterated the importance of the plasticizing effect of CO_2_ on the pressure–permeation rate relationship for glassy polymers.

Although the rate of physical ageing reaches a maximum at temperatures just below the Tg of the polymer [102,103] in more recent studies Xia et al. [104] have shown that the effect can be quite dramatic even at lower temperatures, as a reflection of the involvement of β relaxations as a driving force for molecular packing. Rapidly quenched films made from a proprietary polyimide with a Tg in the region of 310 °C were found to undergo a reduction in gas permeation rate even after only 100 h ageing at 35 °C. Authors found that thin films (136 nm) have intrinsically higher *FFV* than thick films (12 µm) and are more susceptible to physical ageing. Although the rapid physical ageing behaviour was attributed the plasticizing action of the CO_2_ diffusant, this is not a necessary requirement [105,106,107].

Other workers [108,109] have found that the permeation selectivity of ternary gas mixtures containing toluene/CO_2_/CH_4_ and H_2_S/CO_2_/CH_4_ arising from plasticization effects occurred only at very high feed pressure. This anomaly was attributed to the antiplasticization caused by the strong interaction of toluene and H_2_S with the phenylene groups of the polyimide membrane. However, the influential role of pressure cannot be ignored insofar as it can affect permeability through a non-linear increase in solubility.

Although the relationship between permeability (P), solubility (S) and diffusivity (D); i.e., P = S · D, as not received a great deal of attention in the literature, studies concerned with antiplasticization for mixtures of caffeine and PET Burgess et al. [110] have revealed a reduction in permeability expressed as an increase in “barrier improvement factor” (equivalent to the concept of APR, defined earlier). This was found to occur alongside to a decrease in both diffusivity and solubility. From these observations it can be inferred that strong molecular attractions between the diffusant molecules and plasticizers (e.g., H-bond types) can produce a large reduction in permeability. This is demonstrated by the diagrams in Figure 8 for the variation of permeability values for the diffusion of H_2_ and CO at 27 °C through PVC films containing different amounts of TCP plasticizer. The two diagrams display a similarity with variation of free volume with plasticizer concentration shown in Figure 3, which is expected from the relationship between diffusion properties expressed in Equations, (11), (20) and (21).

Moreover, the diagrams in Figure 8 show that the cross-over from antiplasticization to plasticization (T_(AP)_) occurs at concentrations around 25 wt.% TCP for both CO and H_2_. Since these plots indicate that the T _(AP)_ was not affected by the nature of the diffusant it can be deduced that neither of the two gases had a significant plasticizing effect on PVC.

It should also be noted that within the plasticization regime, say at around 40 wt.% TCP content, the permeability values for H_2_ were about 10 times higher than for CO. However, the discrepancy in permeability increased to 100 times within the antiplasticization threshold concentration. A possible explanation is that whereas at low plasticizer concentration the diffusant molecules must penetrate a molecularly homogeneous “bound plasticizer”/PVC phase, at higher concentrations (above 25 wt.% TCP) the interconnected plasticizer clusters provide an easier path for the permeation of either gas.

Within the context of this discussion it is worth noting that the cross-over from antiplasticization to plasticization for glycerol/zein mixtures with respect to oxygen permeability was observed to take place at around 10 wt.% [111] and was attributed to a decrease in the local dipolar relaxation of the amorphous zein matrix. Within the plasticization regime (20 wt.% glycerol), on the other hand, the authors have indicated that glycerol increases the local dipolar relaxation. atomic force microscope (AFM) images have indicated that glycerol has produced “an aggregation of zein complexes”, which may be associated with the presence of 30% mixed amino acids. Such observations highlight the difficulty of predicting the behaviour of plasticizers for systems consisting of complex mixtures, where the effect on properties may result from selective interactions with low molecular weight components rather than the polymer.

#### 3.2.2. Sorption/Desorption and Stability Aspects

Controlled sorption and release properties through antiplasticization have been reported by several authors [112,113,114,115]. The work of Wang et al. [33] (discussed earlier) has revealed an expected increase in water absorption with increasing atmospheric humidity with a minimum at 10 wt.% urea content due to densification (antiplasticization), resulting from enhanced molecular packing of urea and water molecules between starch molecules. The threshold concentration for plasticization was reached at around 20 wt.% urea. The authors suggested that H-bonding associations of starch and urea are formed at around 10 wt.% urea content, evidenced by the shift from 927.8 to 935.5 cm^−1^ absorption for the C–O–C band vibration in the starch. In the latter case the starch/urea bonds are stronger than between urea and water. The observed rapid rise in water uptake at above 20 wt.% urea content was consistent with the model of the plasticization state consisting of continuous clusters, which provides favourable pathways for the penetration of water.

From these considerations and the data derived from the work of Rezus and Bakker [116] on the interactions of water and urea it is possible to obtain a schematic representation of the morphological state of starch containing both urea and water linked by H-bonds, shown in Figure 6, as an elaboration of the illustrations in the report by Wang et al. [33] and shows the incipient formation of clusters of plasticizing species in the transition from antiplasticization, as described by the plot in Figure 3 for this particular system.

Similar structures have been put forward by Van der Sman [117] for mixtures of polysaccharide and polyols where it was shown that phase separation could take place with the addition of water when the conditions for antiplasticization/plasticization threshold are reached.

The molecular structures in Figure 9 show the strong H-bonds between the NH groups of urea with the OH groups in starch and water, alongside the weaker H-bonds between OH groups in carbohydrate chains and water. The data suggest that antiplasticization arise from the strong H-bonds between urea and starch through the likely formation of monomolecular urea–water–urea associations [33]. A further increase in water absorption would not destroy these associations but would provide an increasing amount of weakly bonded assemblies of water molecules close to the carbohydrate chains. At high urea concentrations clusters consisting of a small number of water–urea associations begin to appear and continue to grow in number, leading to the conditions for the transition from antiplasticization to plasticization, in accordance with the principle outlined in Section 2.1.3.

The possibility of producing strong internal association between plasticizers and other auxiliary components has been explored as a means of controlling the release of drugs in pharmaceutical products. Lodagekar et al. [118] have explored the glass forming potential of valsartan for the development of a therapeutically active drug–drug co-amorphous system and have demonstrated that the antiplasticization activity of valsartan played a dominant role. A co-amorphous system with higher valsartan content was found to provide significantly higher dissolution benefits and stability under accelerated conditions for one month. Charmathy and Pinal [119], on the other hand, have used starch containing minor quantities of sorbitol to control the release of theophylline (30 wt.%) through sorption of water from mixtures produced by hot melt extrusion (HME). The rate of release of theophylline was found to increase considerably with increasing sorbitol content above 7.5 wt.% in the HME mixture. Plots of the t50 and t90 values (denoting the time required to reach respectively 50% and 90% release of theophylline) displayed a minimum at 2.5 wt.% sorbitol concentration, followed by a maximum at 7.5 wt.% and then by a monotonic reduction up to 30 wt.% sorbitol content. The authors have attributed the behaviour at very low sorbitol content to an antiplasticization effect of water on starch and, possibly, also to interactions of water–sorbitol.

Densification brought about through the reduction in the fragility of organic molecular glasses has been exploited widely for preserving and maintaining the activity of foods, biological tissues, vaccines, organs, proteins and antibodies. Cicerone and other workers [120,121] have demonstrated that the addition of small quantities of a low-Tg plasticizer (such as glycerol) to a bioprotective glass (such as trehalose) enhances the stability of proteins (enzymes) sequestered within the glass matrix through antiplasticization. These authors have suggested the stability imparted to the protein arises from an antiplasticization effect resulting from the suppression of short-length scale relaxations (i.e., β relaxations) of the glass. Similar conclusions are derived from dielectric spectroscopy studies by Chokshi et al. [122] on the drug indomethacin (INM) dispersed within a multipolymer mixture consisting of solid solutions with Eudragit EPO (rubbery terpolymer of methyl methacrylate, butyl methacrylate and dimethylaminoethylmethacrylate), poly(vinyl pyrrolidone-vinyl acetate) and polyvinyl pyrrolidone homopolymer. The resulting stabilization of the amorphous indomethacin was attributed to strong intermolecular interactions, which suppressed the crystallization of the INM drug through an antiplasticization effect.

#### 3.2.3. The Role of Water in Antiplasticization

The food industry has exploited for a long time the antiplasticization/plasticization transition caused by the absorption of water for converting the original “crispness” of sugary and carbohydrate products (antiplasticization at low water content) to soft matter through the absorption of large quantities of water (plasticization). Labuza et al. [123] have used the concept of brittle/ductile transition to determine the conditions, both in terms of moisture content and temperature. Pittia et al. [124] have found an antiplasticization effect of water in coffee beans at low concentrations from observations of an increase in the compressive strength. This was correlated with a corresponding jump towards longer relaxation times observed on the T_2_ relaxagrams obtained from NMR examinations. Similar effects were noted by Farroni et al. [125] on studies carried out on cornflakes.

In the preceding sections it was shown that the absorption of water could have a large effect on the properties of plasticizer/polymer mixtures even in cross-linked systems due to its capacity of forming strong H-bonding associations and to the intrinsic small size. Accordingly, Wu and Xu [126] have reported a maximum in the measured density and a corresponding minimum in the estimated fractional free volume only at very low water content (1.3 wt.%) for a bisphenol-A epoxy resin cured with isophorone diamine, which is a characteristic feature of antiplasticization.

Ubbink [127] has carried out molecular dynamic simulations for amorphous carbohydrate/water mixtures, which have indicated that the “hole” size within the glassy state decreases at low water contents (antiplasticization regime) as a result of water becoming bound on the polymer chains and have stated that at higher water concentrations are formed domains larger than the free holes. This suggests that the formation interconnected clusters results from weak polymer–plasticizer interactions, which can be inferred also from the discussion about the events identified in Figure 9.

Water has also a unique role for its ability to increase the Tg of polymers at low concentrations even in systems that do not possess many groups with strong H-bonding capability. An example is the increase in Tg of a commercial soft polymer material consisting of a random terpolymer of methylmethacrylate, butyl methacrylate and dimethylaminoethyl methacrylate, known as Eutragit E 100 [128] through sorption of moisture from the environment. An increase in Tg from 34 to 42 °C was observed at 6% RH in correspondence with a substantial reduction in the water permeation rate at the same condition.

An example of antiplasticization has also been identified for a molecular glass by Ruiz et al. [129] where it was shown that addition of water to prilocaine, an active pharmaceutical ingredient, has the same effect as that of an applied pressure, resulting in an increase of Tg. The antiplasticization effect was ascribed to the formation of prilocaine-H_2_O dimers or complexes with enhanced hydrogen bonding interactions, which is a similar situation to the interactions of water with urea described earlier.

## 4. Peculiarities in the Interpretation of Antiplasticization Phenomena

### 4.1. The Glass Transition Temperature Anomalies

As indicated in the preceding section on the role of water, some authors have interpreted the term “antiplasticization” as the opposite effect to plasticization with respect to the change in glass transition temperature. Luk et al. [130], for instance, have studied the effect of adding up to 8% amylose complexing fatty acids (CFA), such as linoleic and oleic acids, on the glass transition temperature of cassava starch (CS) with moisture content up to 35% (dry basis). For water contents below 15 wt.% a large increase in Tg was observed with the addition of CFA with a maximum at 2 wt.%, which diminished with increasing water content. The authors have attributed this peculiar antiplasticization effect to the formation of amylose–lipid inclusion complexes, which act as “physical cross-links” for the CS–water associations. Garcia et al. [131], on the other hand, have used the term antiplasticization to describe the increase in Tg of an amorphous aromatic/aliphatic polyamide (Trogamid T–5000) with the addition of minor amounts of a higher Tg auxiliary (polyvinyl phenol). However, this is an expected behaviour of miscible polymer blends arising from the difference in the Tg between the two polymer components and, therefore, it must be examined in a different light than an antiplasticization phenomena.

A similar outcome has arisen from dielectric spectroscopy studies on mixtures of polybutadiene and mineral oil by Casalini et al. [132]. These authors have observed a shift of the α-relaxation spectrum to lower frequencies and have concluded that such an “antiplasticization behaviour” can only arise if the Tg of the oil is higher than the Tg of the polymer. In this case, the authors have pointed out that they were not able to obtain an experimental verification of the Tg of the oil owing to its crystallization in cooling the sample to low temperatures. It is worth noting also that the presence of oil in the mixture was found to increase the magnitude of the α-relaxation time, without any effect on the β-relaxations. Moreover, neither the shape of the α-relaxation function nor its temperature dependence was influenced by the presence of oil. Consequently, such a behaviour cannot be associated with antiplasticization.

In a recent report Andrews et al. [133] have used the term antiplasticization to describe as a phenomenon that brings about a strong deviation from the Gordon Taylor equation. In the related work the nonsteroidal antiandrogen Bicalutamide (BL) was mixed with water soluble polyvinylpyrrolidone (PVP) by hot melt extrusion (HME), using 10 wt.% triethylcitrate as the “processing plasticizer”. The mixtures exhibited a single Tg value, which were significantly higher than those calculated using the Gordon–Taylor equation and the behaviour was attributed to strong intermolecular interactions between BL and PVP.

### 4.2. Implications of Data on Molecular Glasses

Contrary to the behaviour of polymer glasses discussed in relation to mechanical properties (Section 3.1) dielectric relaxation studies on organic molecular (low molecular weight) glasses have revealed the existence of a singular relationship for the temperature/plasticizer concentration to identify the threshold conditions for the cross-over from antiplasticization to plasticization. This is exemplified by the data and observation reported from several studies carried out on mixtures of trehalose (a dimer of glucose) and glycerol. In dielectric relaxation studies carried out over a wide range and frequency by Obrzut et al. [134] and Curtis et al. [135] the “relaxation time of the system” ( τ ) for the JG β-relaxations was estimated from the complex permittivity data in terms of the distribution function of relaxation times and a regularization technique [136].
(23)$$\frac{\varepsilon \;\left(\omega \right)-\varepsilon \;\left(\infty \right)}{\Delta \varepsilon }=\int^{\;}\frac{1}{1+i\;\omega \tau }\;G\;(\ln\;\tau )\;d\;(\ln\;\tau )$$
where ε (ω) is the permittivity at the measurement frequency, G (ln τ) is the logarithm distribution function of relaxation times and Δε is the difference between relaxed (ε_o_) and unrelaxed (ε∞) permittivity. The ratio of the relaxation time of the mixture to that of trehalose, τ_mix_/τ_th_, (conceptually analogous to the *APR* factor for application related properties) was used to characterize the extent of antiplasticization. In Figure 7 are shown the variation of τ_mix_/ τ_th_ with increasing glycerol concentration and a plot of “Threshold temperature” (T_A/P_) against “Weight fraction of glycerol (ω _glycerol_)”, obtained from the intersection at τ_mix_/τ_th_ = 1

The salient feature of the diagrams in Figure 10 (left) is the very strong antiplasticization character of the trehalose/glycerol mixtures, indicated by the large peak τ_mix_/τ_th_ values. The data for the threshold temperature in Figure 10 (right) show that the cross-over from antiplasticization occurred at a temperature well below the Tg of trehalose (115 °C) even at very low plasticizer concentrations. The data are in good agreement with those reported more recently for the same binary system by Ubbink [137] with respect to the rapid reduction in T_A/P_ at glycerol concentrations greater than 30 wt.%. Similarly, Anopchenko et al. [138] have used the relaxation time ratio (τ_mix_/τ_th_) as a measure of the extent of antiplasticization and have obtained similar results described in Figure 10, using the term “critical plasticization” concentration to characterize the cross-over from antiplasticization to plasticization.

It is worth noting that Riggleman and de Pablo [139] have performed molecular dynamics simulations for trehalose/glycerol mixtures, which predict an increase in density at 5 wt.% concentration at all temperatures below 27 °C. For the same system Ubbink [22] have reported a maximum in Debye–Waller factor at the same glycerol concentration from neutron scattering measurements. This is consistent also with the data obtained for mixtures with sucrose [140], which showed a maximum in the stiffness at 5 wt.% glycerol.

From these studies it is not possible to deduce whether the divergence from the dual cross-over behaviour discussed for polymer glasses in relation to mechanical properties arises from molecular size discrepancy or the nature of the excitation for the induced relaxations. It is possible, however, that measurements have not been carried out at sufficiently low temperatures to reveal the conditions for a reversion to plasticization behaviour, (i.e., τ _mix_/τ_th_ < 1) at some critical range of plasticizer concentration.

Weng and Elliott [141] have used dynamic mechanical tests (DMA) on trehalose/glycerol mixtures to obtain mechanical spectra for the construction of master curves using the Williams-Landel-Ferry (WLF) shift factor **a_T_** method for the time/temperature superposition principle. In dynamic mechanical spectroscopy this can be expressed in terms of a horizontal shift for the variation of the loss modulus with a frequency over a wide range of temperatures, i.e., E″(ω, T) = E″(**a_T_** ω, Tr), where Tr is an arbitrary reference temperature. This gives rise to an equivalent shift in the α-relaxation time using the E″ peak value along the frequency axis, so that *τ* = **a_T_**
*τ*_r_ and *τ*_r_ = 1/ω_max_ according to the linear viscoelastic theory. Through substitutions it is possible to obtain estimates for the fragility index *m* from the constants C_1_ and C_2_ of the WLF equation, log *τ/τ*_g_ = −C_1_(T−Tg)/C_2_ + (T−Tg) where *τ*_g_ is the relaxation time at Tg. This results in the expression *m* = C_1_ Tg/C_2_, where Tg is in K unit.

The plot shown as an inset for Figure 10 (right) show a minimum for the *m* values obtained by this method at around 90–92 wt.% and a maximum at around 85 wt.% trehalose content, while the composition dependence of Tg was found to follow a Gordon–Taylor-like relationship. Furthermore, the authors have shown that the line joining the two points for *m* at the higher glycerol concentration with the value of *m* for pure trehalose coincides with the data obtained by the additivity rule for the glass fragility of the mixture. The *m* value for glycerol was obtained by molecular dynamics simulations.

There are significant merits in the use of the fragility factor for determining the deviation of the relaxation time from the additivity rule for mixtures, in so far as it considers events at Tg rather than at an arbitrary temperature, as in the case of the specific volume in Table 2. Nonetheless, the possibility of inducing a reversion to plasticization at lower temperatures relies on the likelihood of an increase in the related activation energy of molecular relaxations in the β and γ transitions region, as indicated by the data in the report by Maeda et al. [80].

### 4.3. Antiplasticization in Heterogeneous Polymer Materials

#### 4.3.1. Polymer Blends

Moraru et al. [22] have found a significant antiplasticization effect, manifested as an increase in both shear and tensile storage modulus, in polyol plasticized mixtures with meat–starch (carbohydrate/protein blend) resulting from the addition of a very small amount of glycerol (2 wt.%) and through the absorption of small amounts of moisture. Liu et al. [142] have reported a similar effect at approximately the same glycerol content (2.5 wt.%) for films produced from blends of starch and chitosan. The authors have shown that the addition of glycerol promotes interactions between chitosan, starch and glycerol through hydrogen bonding, evidenced by a shift to high wave numbers of the main peaks located at 3328, 2927 and 1638 cm^−1^ of starch and chitosan films without glycerol.

Saedi et al. [143] have studied the permeation of CH_4_/CO_2_ mixtures though membranes produced from blends of polyethersolphones (PES) with minor amounts of polyetherimide (PEI). A minimum was found in the variation of permeability and a maximum in the selectivity at a weight ratio of PES/PEI equal to 98/2. At this composition ratio the two polymers were found to be fully miscible, exhibiting only one Tg occurring at a temperature between the Tg values for the single components.

A very intriguing aspect of antiplasticization was revealed by Modesti et al. [144] in studies on the permeation of water vapour through membranes made from a polyester (PEst) based polyurethane (PU), which is a thermoplastic elastomer consisting of rigid PU domains dispersed in a rubbery PEst matrix.

The authors have used the exponential Long’s equation [145] for the relationship between diffusion coefficient (D_eff_) and volumetric water fraction sorbed by the membrane (φ), which is based on free volumes theory for a linear increase due to plasticization, i.e.,
(24)$$D_{\mathrm{eff}}=\mathrm{Do}\;e^{\gamma \varphi }\;\;$$
where Do is the pre-exponential term and γ is a factor that characterizes the plasticization efficiency of the diffusant.

Fitting the results to the above equation has produced a negative value for γ , which has been associated with antiplasticization due to clustering of water. From a thermal analysis using modulated differential scanning calorimetry (DSC) the authors were able to identify the fraction of water “bound” on the polymer chains (PU segments) at low water fractions and water “cluster” at higher water contents.

From an analysis of data on the antiplasticization effects of water on mixtures of starch and polyols Van der Sman [117] has shown that phase separation can take place at a specific composition ratio even in the glassy state for polyol concentration above the antiplasticization/plasticization threshold condition. This observation extends the sequence of events and change of physical state taking place with increase plasticizer content in the following order: antiplasticization (molecular-scale homogeneity) → plasticization (nanoscale domain heterogeneity) → phase separation (micron-scale domain growth).

In relation to phase heterogeneity Ward and Koros [146] and also Kamaruddin [147] have specifically probed the effects of antiplasticization and physical ageing on mixed matrix membranes (MMM) based on cross-linked polyimides. Some intriguing peculiarities were observed for the effect of toluene contamination in the CO_2_/CH_4_ mixture. A higher reduction in permeability was observed at toluene contamination of 1000 ppm than at 500 ppm, which was attributed to a combination of dual mode sorption and antiplasticization effects.

#### 4.3.2. Polymer Matrix Composites and Nanocomposites

A typical antiplasticization behaviour was revealed by Rasoldier et al. [148] with respect to the thermal oxidative degradation of epoxy-matrix carbon-fibre composites. The authors observed that the products of chain scission reactions gave rise to a reduction in the glass transition temperature and a corresponding increase in modulus in the temperature range between the α and β transitions with two cross-over points. Vassileva and Friederich [149] have observed suppression in the β relaxations and an increase in T_β_ with the incorporation of alumina nanoparticles into an amine cured epoxy resin, which they have attributed to an antiplasticization feature resulting from densification of the resin matrix around the dispersed nanoparticle. A similar effect on β relaxations created by the presence of a filler (insoluble high-surface energy component of a mixture) was recently observed by Mascia et al. [150] in studies on the thermal transition by DSC analysis of lactobionic acid (organic molecular glass). The relaxation peak below the Tg was shifted to higher temperatures and became more pronounced through physical ageing, which can be taken as a confirmation that the antiplasticization effect observed in a “thermoset” polymer glass can be attributed to the presence of “free” low-MW species, as also indicated in the findings of Rasoldier et al. [148]. This interpretation is supported by the work of Tang et al. [151], in which was found a maximum in tensile strength and a minimum in water vapour transmission at 5 wt.% glycerol content for a starch-based composite containing 6 wt.% montmorillonite nanoclay. However, these effects were not related to β relaxations and, therefore, must be regarded as a mere manifestation of the expected behaviour of “composites”.

### 4.4. Unusual Feature of Antiplasticization

An unusual behaviour, which was attributed to antiplasticization, has been reported by Michelis et al. [152] in studies of the effects of ibuprofen as a plasticizer for a proprietary acrylic pressure sensitive adhesive, which has a Tg in the region of −22 °C. An increase in both storage and loss modulus, as well as an augmentation of the “probe tack” stress value measured at 21 °C, was observed at 1 wt.% level of addition. The striking anomaly in the findings of this work was in respect to the antiplasticization effect (expressed as in increase in stiffness), which was observed within the rubbery state at a temperature well above the Tg. Since the authors were not able to reveal the presence of crystals by both polarizing microscopy and DSC analysis, the observed anomaly can only be attributed to “physical cross-links” resulting from strong H-bonding interactions between the COOH in ibuprofen and highly polar groups in along the acrylic polymer chains, which are required to achieve a high level of adhesion.

A similar intriguing behaviour has been observed for the effect of water on a mixture of cassava starch with 6 wt.% corn oil [153], which showed an increase in the rubbery plateau modulus when the water content increased to 19 wt.% and higher.

## 5. Concluding Remarks and Future Perspectives

The analysis of the work published over the last several decades has brought to light useful findings that have made it possible to construct a model for polymer/plasticizer interactions within the context of antiplasticization and related morphological features, as well as valuable insights for applications. The main findings are summarised in Table 4.

The analysis has identified the deviation from the additivity rule for free volumes and fragility factor as the common basis for the phenomenological events leading to antiplasticization, At the same time, it has not been possible to demonstrate the universal existence of two “threshold” temperatures for the antiplasticization/plasticization cross-over for diffusivity and permeability, which was identified with respect to mechanical properties. There are two possible reasons for this deficiency: (a) the difficulty of obtaining the required data from experiments carried out at sufficiently low temperatures and (b) the lack of a strong “benefit pull” for applications, such as the fields of membranes and drug delivery. Another weakness of existing theories on antiplasticization is the lack of considerations regarding the viscoelastic nature and yielding behaviour of polymers [9,154].

Up to present time molecular dynamics and LCL studies have not considered the dilatational effects of hydrostatic stresses that bring about failure through crazing, microcavitation and fracture [12,20,83,85,155]. Another shortcoming of present theoretical studies is the lack of regard for the vanishing of antiplasticization at low temperature, which could be overcome through better understanding of changes in the activation energy for relaxations in the β and γ transitions region.

Another aspect that requires attention in theoretical models for antiplasticization is the cooperative effect of physical ageing. The embrittlement behaviour observed on various plasticized chitosan films by Suyatma et al. [156], for instance, suggests the threshold plasticizer concentration for the cross-over to plasticization is higher for samples that have been subjected to physical ageing.

Regarding future developments in material compositions it is possible that deep eutectic solvents (DES) may feature more prominently in experimental work. The use of urea-acetamide as a DES in polymer compositions could attract some attention as it is expected to be a more effective plasticizer than the corresponding single crystalline components, owing to its glassy nature with a low Tg (−30 °C) [157]. This hypothesis is supported by the findings of Ma and Yu [158] from a study on plasticization of starch by binary plasticizer combinations of single components of urea with formamide and urea with acetamide for the production or rubbery polymer compositions, where it was observed that an effective plasticization system was obtained only at some specific plasticizer ratios, corresponding to conditions that prevent the segregation of urea. A paper published by Sousa et al. [159] using choline chloride-urea DES as a solvent and permanent plasticizer to produce agar films has already indicated the viability of these systems. Similarly, the incorporation of ionic liquids in thermosetting polymers [160] has also been found to produce a typical antiplasticization behaviour, i.e., reduction in Tg and increase in modulus at temperatures within the glassy state. These could prove to be particularly valuable for cold-cured adhesives and composites in view of the advantages ensuing from the reduction in viscosity and Tg, which are expected to bring about an increase in both rate of hardening and final cross-linking density [155]. This suggestion is validated by observations on cold-cured epoxy-siloxane hybrid systems, showing that a reduction in the rate of the evolution of Tg during the first stage of curing has resulted in higher Tg values and enhanced mechanical properties in the final stage [161].

Antiplasticization of oriented products, films and fibres, is another area for further exploitation [162,163,164]. It should be noted that the occurrence of antiplasticization has been revealed with respect to mechanical properties of electrospun fibre mats produced from zein mixtures with glycerol as the plasticizer. Moreover, in a recent report Mascia et al. [165] have widened the horizon for the use of plasticizers in the electrospinning of zein fibres in a manner that could provide new opportunities for exploring the concept of antiplasticization to produce antibacterial fibre mats.

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
