# Peer review of "Antiplasticization of Polymer Materials: Structural Aspects and Effects on Mechanical and Diffusion-Controlled Properties"

_polymers, 2020, doi:10.3390/polym12040769_

Round 1
Reviewer 1 Report
In the review "Antiplasticization of polymer materials: structural aspects and effects on applications related properties", authors present a comprehensive overview of many of the studies relating to plasticization and antiplastisization of polymeric materials. Although, authors refer to a lot of previous work in the field, I am not convinced as to how this work is different from previous reviews in the fields and what impact does it add to the already existing body of work. Based on this, I do not recommend the publication of this article in it's current form.
Below are few comments I have.
- Although there is a brief reference to work done Riggleman et.al., authors can add a whole new section highlighting the contribution of modelling and simulations to this field. I believe this might be something that will add impact to this article.
- In lines 84-87 authors talk about fragility playing a role in antiplasticization. This is another new concept to the field and my recommendation will be to talk a lot more about fragility and how it impacts glass formation behavior of polymers (can refer to extensive work done by David S. Simmons and his research group)
- I like the schematic S1, but I am unable to understand the complete logic behind construction of such a schematic. It would be productive, If the authors can talk a little more about it.
- I found there are places where the is a lot of inadequate references. Eg. Table 1, lines 127-129, Figure 2(a) etc.
- Section 2.1.2. about Polymer/plasticizer interactions by FTIR, I did not find a particular use of going into details when the conclusion was this method showed no indications in the literature FTIR examinations are capable to determine unequivocally whether the polymer/plasticizer interactions are related to antiplasticization.
- Also, the formatting of the references is not adequate. Eg. 37
Author Response
Authors Response attached

Reviewer 2 Report
The critical comments are as follows:
1. The title is not as per work contents and needs modifications. The term “effects on applications related properties” is vague.
2. The abstract is poorly presented. It doesn’t deduce the justification of the literature gap. Normally an abstract, for a review, should state briefly the purpose of the study undertaken and meaningful conclusions. Herein, I would like to see the major findings and how they are addressing the left behind research gaps and covering current challenges
3. At most, the authors have used vague sentences. The level of English used is not up to the journal standard. Throughout the manuscript, the level of English used is not up to the standard of the journal. The sentences are long and badly worded with repetitive words. Please consider breaking longer sentences into smaller fragments for easy understanding. Authors are advised to seek help from a native English speaker.
4. What was the review methodology? How the literature was analyzed? What was the inclusion/exclusion criteria?
5. More Figures and tables should be added to each section/subsection.
6. L96: “S.1 Schematic outline” is superficial. Reconstruct the Figure with a detailed mechanistic approach and elaborate.
7. L120: Table 1 is poorly organized without any supporting reference. What was the data source?
8. Table 2 and so on – the same comment as above for Table 1. Revisit and update.
9. Figure 2 – what is the source data? It is authors responsibility to avoid copyright issues. I wonder if the permission has been taken from the copyright holder sources. If so, the statement like Reproduced/Adapted with permission from a respective source… should be declared.
10. The entire conclusion is full of generality and based on already reported literature. A complete rewrite is required from the authors' own viewpoint for future considerations.
Author Response
Authors response attached

Round 2
Reviewer 2 Report
The revised version reads well. Authors have addressed all the comments raised in the last review. This manuscript can now be accepted for publication.